# Nanohybrid Assemblies of Porphyrin and Au_10_ Cluster Nanoparticles

**DOI:** 10.3390/nano9071026

**Published:** 2019-07-18

**Authors:** Mariachiara Trapani, Maria Angela Castriciano, Andrea Romeo, Giovanna De Luca, Nelson Machado, Barry D. Howes, Giulietta Smulevich, Luigi Monsù Scolaro

**Affiliations:** 1CNR-ISMN, Istituto per lo Studio dei Materiali Nanostrutturati c/o Dipartimento di Scienze Chimiche, Biologiche, Farmaceutiche ed Ambientali, University of Messina V. le F. Stagno D’Alcontres, 3198166 Messina, Italy; 2Dipartimento di Scienze Chimiche, Biologiche, Farmaceutiche ed Ambientali and C.I.R.C.M.S.B., University of Messina V. le F. Stagno D’Alcontres, 3198166 Messina, Italy; 3Dipartimento di Chimica “Ugo Schiff”, Università di Firenze, Via della Lastruccia 3-13, 50019 Sesto Fiorentino (Fi), Italy

**Keywords:** gold clusters, plating, porphyrin, chirality, SERS

## Abstract

The interaction between gold sub-nanometer clusters composed of ten atoms (Au_10_) and tetrakis(4-sulfonatophenyl)porphyrin (TPPS) was investigated through various spectroscopic techniques. Under mild acidic conditions, the formation, in aqueous solutions, of nanohybrid assemblies of porphyrin J-aggregates and Au_10_ cluster nanoparticles was observed. This supramolecular system tends to spontaneously cover glass substrates with a co-deposit of gold nanoclusters and porphyrin nanoaggregates, which exhibit circular dichroism (CD) spectra reflecting the enantiomorphism of histidine used as capping and reducing agent. The morphology of nanohybrid assemblies onto a glass surface was revealed by atomic force microscopy (AFM), and showed the concomitant presence of gold nanoparticles with an average size of 130 nm and porphyrin J-aggregates with lengths spanning from 100 to 1000 nm. Surface-enhanced Raman scattering (SERS) was observed for the nanohybrid assemblies.

## 1. Introduction

Gold metal nanoparticles with diameters smaller than ~2 nm have received considerable interest in nanoscience because they can be controlled with atomic precision, and, due to discrete energy levels and molecular-like HOMO–LUMO transitions, exhibit optical features fundamentally different from those of larger nanoparticles [1]. Commonly, the term nanoparticles (NPs) is used to refer to entities with diameters greater than 2 nm and an indefinite structure, whereas defined stoichiometric species with a diameter smaller than 2 nm are described as nanoclusters (NCs) [2]. The knowledge of the exact number of atoms present in gold nanoclusters is a very important aspect, as these clusters constitute the link between atomic and AuNP behavior. In this respect, AuNCs show discrete, size-dependent absorption and fluorescence emission from the UV to the near-IR spectral regions, together with significant quantum yield values [1,3,4]. Nonlinear optical (NLO) properties of gold NCs, such as third-order optical nonlinearity, two photon absorption (TPA), two-photon excited emission, and hyperpolarizabilities have been investigated [5,6,7,8,9]. Furthermore, AuNCs exhibit high photostability and biocompatibility, characteristics advantageous in biomedical applications, which can be improved by refining the synthesis, processing, and surface coating of the NCs [10,11,12]. Surface ligands, such as dendrimers, polymers, proteins, and oligomers play a key role in the stability of AuNCs in solution, and also affect their structure and optical properties [1,3]. In this framework, the use of biomolecules as capping ligands improves biocompatibility, and is considered an efficient way to transfer chirality to nanoclusters [13,14,15]. Indeed, three possible mechanisms for chirality transfer have been reported: (i) chiral growth, due to the formation of a chiral metal core influenced by the presence of chiral ligands, (ii) chiral polarization, due to electronic interaction between an achiral metal core and chiral ligands, and (iii) chiral footprint, due to the arrangement of the ligands on an achiral metal core [16,17]. In this respect, chiral emitting gold nanoclusters, in which the two enantiomers of histidine (His) have been used as both reducing agent and protecting ligands, have been easily synthesized by a one-step approach [18,19]. The presence of suitable ligands bearing –NH_2_, –COOH, or polymerizable substituents open the way to further functionalization of NCs for applications in sensing, bioimaging, and energy transfer [20,21,22]. A central focus throughout materials research is control of the organization of the clusters on nanostructured surfaces, as well as understanding of the growth mechanisms at an atomic scale in order to obtain well defined and uniform architectures [23]. In fact, the quality of the metallic layers has a strong impact on the mechanical, electrical, and optical properties of the films. Several procedures to obtain gold nanostructure thin films have been reported, spanning from sputtering [24], lithographic methods [25], chemical vapor deposition (CVD) [26], electroless deposition (ELD) [27,28], and self-assembly approaches, which involve a variety of substrates [29,30]. In this framework, it has been reported that self-assembly of monodisperse gold colloid particles into monolayers on polymer-coated substrates produces highly reproducible macroscopic surfaces active for surface-enhanced Raman scattering (SERS) [31]. In fact, many spectroscopic studies on the interaction of AuNPs with porphyrins and their aggregates in solution and on surfaces have been reported [32,33,34,35,36]. In this particular area, our interest has focused on the J-aggregates of tetrakis(4-sulfonatophenyl)porphyrin (TPPS) [37,38,39,40,41,42]. These aggregates are formed under acidic conditions and the partially protonated porphyrins are arranged in a lateral stacking geometry, which leads to the occurrence of very peculiar optical properties [43,44]. Due to their manifold applications, various reports have dealt with the immobilization of J-aggregated porphyrins on substrates, highlighting the importance of the deposition step [45]. In particular, microscopic analysis has revealed the presence of differently shaped aggregates, such as rod, and nanotubular structures, depending on the experimental conditions [35,43,46]. Moreover, a detailed study has revealed that porphyrin concentration strongly affects the amount of J-aggregate adsorbed onto glass surfaces, showing a higher number of adsorbed entities at lower porphyrin concentrations [47]. In the past, J-aggregated porphyrin has been exploited for the design of inorganic/organic nanocomposites in combination with gold nanoparticles and spermine [48] or with gold nanorods [36,49]. Herein, we report on the ability of sub-nanometer sized gold clusters (Au_10_) capped with l- or d-histidine to induce the formation of chiral TPPS J-aggregates under rather mild acidic conditions. We describe a simple procedure to co-deposit gold nanoparticles and chiral porphyrin J-aggregates onto glass substrates by simple acidification of an aqueous solution of such Au_10_ clusters. Furthermore, these films have been shown to be active substrates for SERS, as demonstrated by the observation of intensified Raman signals for the co-deposited porphyrin J-aggregates. To the best of our knowledge, this is the first time that gold clusters composed of ten atoms (Au_10_) have been used in combination with porphyrins to build a nanohybrid assembly (Au_10_@Jagg) able to self-organize on a solid that can be potentially used as a SERS-active substrate for the detection of chemically- and biologically-relevant species.

## 2. Materials and Methods

*Chemicals*. Hydrogen tetrachloroaurate(III) hydrate (99.9%) was supplied by Strem Chemicals (Bischheim, France). d- and l-histidine (98%) were obtained from Sigma-Aldrich (Milan, Italy). The 5,10,15,20-tetrakis(4 sulfonatophenyl)porphyrin (TPPS) was purchased from Aldrich Chemicals (Milan, Italy), and its solutions of known concentration were prepared using the extinction coefficient at the Soret maximum (ε = 5.33 × 10^5^ M^−1^ cm^−1^ at λ = 414 nm). Hydrogen peroxide (30%, Sigma Aldrich, Milan, Italy), NaBH_4_ (98%, Aldrich, Milan, Italy), potassium nitrate, and sulfuric (98%), hydrochloric (37%), and nitric (69%) acids (Fluka, Milan, Italy) were used. All the reagents were used without further purification and the solutions were prepared in dust free Milli-Q water (Merck, Darmstadt, Germany).

*Gold cluster synthesis*. Synthesis of gold clusters (Au_10_) was carried out according to a literature procedure [18]. Briefly, a solution of l- or d-histidine (0.1 M, 6 mL) was added to the solution of the metal precursor (HAuCl_4_, 10 mM, 2 mL) under continuous stirring at 25 °C for 2 h. After this time, the solution turned pale yellow and was used in this form.

*Plating procedure.* Small pieces of glass cover slides were thoroughly washed in an acid piranha solution (H_2_SO_4_:H_2_O_2_ 4:1 v/v). After being rinsed in pure water, the slides were vertically immersed in 3 mL of solution containing gold clusters (1.5 mL) and HCl (pH 2.0). The metallic structures were left to grow on the substrates for 24 h at room temperature. Care was taken to remove any excess histidine possibly adsorbed onto the surface by dipping the slides in acidic water (pH 2.0). Finally, the substrates were dried under a gentle nitrogen flow.

*Au_10_@Jagg assemblies*. Glass slides, carefully cleaned according to the procedure described above, were vertically immersed in 3 mL of solution containing gold clusters (1.5 mL), porphyrin (up to a concentration of 5 μM), and HCl (pH 2.0). After an aging time of four days at room temperature, the slides were dried under a gentle nitrogen flow and the excess histidine was removed by quick immersion in water at pH 2.

*Spectroscopic and morphological characterization.* UV–vis spectra were collected on a diode-array spectrophotometer Agilent model 8452, subtracting the spectrum of a clean glass slide. Circular dichroism (CD) spectra were recorded with a Jasco model J-720 spectropolarimeter. Atomic force microscopy (AFM) measurements were performed using a NanoSurf easyScan2 microscope operating in non-contact mode at room temperature, with a resolution of 512 × 512 pixels and a moderate scan rate (1–2 lines/s). Commercial Si-N-type tips (AppNano mod. ACLA) with resonance frequencies of 145–230 kHz were used. Fluorescence emission and resonance light scattering (RLS) experiments were performed on a Jasco mod. FP-750 spectrofluorimeter. A synchronous scan protocol with a right angle geometry was adopted for collecting RLS spectra [50], which were not corrected for the absorption of the samples. Raman and SERS spectra were obtained at room temperature using a Renishaw RM2000 single-grating spectrograph apparatus, equipped with an Ar^+^ laser at 514.5 nm and a near-infrared diode laser at 785 nm. Measurements were made in backscattered geometry using a 50× microscope objective filtered by a notch holographic filter, dispersed by a single grating (1200 lines mm^−1^), into a charge-coupled device (CCD) detector cooled to −70 °C by the Peltier effect. Spatial resolution was 2 mm and the spectral resolution was 3 cm^−1^. Laser power at the sample was in the range 17–185 μW (514.1 nm exc.) and 200 μW–2 mW (785 nm exc.). No sample degradation was observed under these conditions. To improve the signal-to-noise ratio, a number of spectra were accumulated and summed only if no spectral differences were noted. Raman and SERS spectra were calibrated with indene and CCl_4_ as standards. In order to measure Raman intensities, nitrate and sulfate were used as internal standards. Therefore, to this end, porphyrin aggregation was fostered by means of the addition of KNO_3_ (150 mM) and HNO_3_ (32 mM) to the porphyrin solution (5 μM). Due to the overlap of the 993 cm^−1^ band of sulfate and a porphyrin mode, the 1053 cm^−1^ band of NO_3_^−^ was used to normalize the spectra.

## 3. Results

Au_10_ clusters capped with d- or l-histidine were obtained according to a previously reported procedure [18]. The imidazole group of histidine plays a key role in AuNCs formation, as it serves as both a reducing agent and a protecting ligand. The clusters showed an absorption edge at around 450 nm that rose very steeply below 320 nm (full line spectrum, inset of Figure 1), together with the typical fluorescence emission centered at 500 nm, as previously reported [19]. No significant spectroscopic differences were observed for the Au_10_ sample stabilized by the two different histidine enantiomers. The chiroptical properties of histidine AuNCs were investigated by circular dichroism spectroscopy, confirming that the chirality of the histidine ligands was transferred to the metal core [19]. When TPPS (5 μM) was added to the AuNCs solution, the electronic spectrum for both histidine enantiomers showed the presence of a porphyrin Soret band centered at 418 nm and four bands located in the visible region at 517, 554, 587, and 642 nm (red spectrum, Figure 1). The positions of both the B and visible bands were bathochromically shifted compared to those of the free base TPPS in aqueous solution in the presence of histidine [51].

Furthermore, the CD spectra of the two enantiomeric forms showed a signal in the porphyrin absorption region, indicating the occurrence of an asymmetrical perturbation resulting from the chiral AuNCs. The CD spectra (inset of Figure 2) performed for l- and d-His-AuNCs enantiomers showed signals in the porphyrin region that were correlated with the enantiomorphism of the stereocenter present on the AuNCs protecting ligand, with a negative and positive bisignate Cotton effect for l- and d-histidine, respectively. This suggests that chiral information was successfully expressed on the porphyrin chromophore at the supramolecular level. The low induced CD intensity was in line with the monomeric nature of the chromophore. Recently, it has been reported that no interaction between TPPS and histidine has been detected in aqueous solution, mainly due to electrostatic repulsions between the negatively charged porphyrins and the amino acids [51].

This was confirmed by the absence of any CD signal induction in the porphyrin region. Nevertheless, as has been observed for other systems, the absorption band shifted and the induced CD signal could be ascribable to electrostatic interactions or hydrogen bonds between the negative sulfonated groups of the macrocycle and the protonated amino groups of histidine, and/or to the localization of the porphyrin in a hydrophobic microenvironment due to the histidine surrounding the metal nanostructures [52,53,54,55]. Moreover, the formation of positively charged histidine oligomers with Cl^−^ as a counter anion has been reported [18]. Therefore, it also cannot be excluded that electrostatic interactions between the positively charged histidine oligomers and the negatively charged sulphonate groups present in the periphery of the macrocycle were the origin of the spectral variations. It should be mentioned that, upon addition of the positively charged tetra N-methylpyridinium porphyrin (H_2_TMPyP^4+^) to the Au_10_ solution, the Soret band red-shifted to 427 nm with respect to the free base (422 nm) (ESI, red spectrum, Appendix A). This shift can be explained by an interaction of the carboxylic groups of histidine with the positively charged N-methylpyridinium groups at the meso position of the macrocycle. Since histidine Au_10_ clusters can be successfully prepared over a wide pH range (pH 2–12) [18], we decided to foster porphyrin aggregation by lowering the pH. When HCl (pH 2) was added to the TPPS Au_10_ cluster solution, the UV–vis spectra showed an almost instantaneous formation of diacid TPPS with a Soret band at 435 nm (green spectrum, Figure 1), which slowly interconverted into J-aggregates characterized by a narrow peak located at 492 nm. Furthermore, the electronic spectrum at the end of the aggregation process showed a very broad band at around 550 nm not ascribable to porphyrin features (blue spectrum, Figure 1). For comparison, a control experiment was performed in the same conditions in the absence of porphyrin. In this case, acidification of the Au_10_ cluster aqueous solution instantaneously induced the formation of a new band at 400 nm (dashed line spectrum, inset Figure 1) that broadened and red-shifted within 1 h to around 550 nm (ESI, red spectrum, Appendix A), and eventually it led to a general increase of the baseline within 24 h (dotted line, inset Figure 1). This effect is ascribable to the growth of AuNcs to form AuNPs, which show the typical plasmon resonance band [56]. The process was accompanied by a color variation of the solution from pale yellow to pink. It is reasonable to expect that upon acidification of the Au_10_ cluster solution, larger metallic structures were produced by aggregation of the clusters, which resulted from hydrogen bond formation between the carboxylic and protonated amino groups of adjacent units of the histidine residues covering the gold surface, as already reported for AuNPs stabilized by histidine [57]. In fact, in the case of heterocycle amines such as pyrrole or tryptophan, the occurrence of macroscopic segregation of the polymer due to the amine oxidation process and the metal phase has been reported as a function of the heterocycle amines/gold stoichiometric ratio [58]. After removing the solution from the cuvette, a purple deposit was observed on the quartz surface (ESI, Appendix A). The corresponding electronic absorption spectrum, obtained from washing the cuvette with water, exhibited a broad band that extended from ca. 550 nm to higher wavelengths very similar to that observed 24 h after HCl addition (inset Figure 1). This effect is ascribable to the adhesion of the metal nanoparticles to the quartz surface. A similar experiment was carried out at the same pH value, reducing the amount of gold clusters. In this case, the UV–vis spectra evolved from the initial 400 nm band to a broad feature extending from ca. 550 nm to the near-IR, but neither the 530 nm band due to AuNPs nor flocculated material were detected (ESI, Appendix A). This suggests the direct growth of small gold clusters into larger entities with nucleation on the surface, without the formation of larger metal colloidal suspensions in solution. This observation may result from slower kinetics related to a lower amount of Au_10_ clusters, thus improving the homogeneity of the deposited material onto the quartz surface. At a higher Au_10_ load, the formation of solid material could be ascribed to a faster growth process causing the formation of much larger metal structures that become unstable in solution and eventually precipitate. It is noteworthy that no metallization of the cuvette wall occured when Au nanoparticles synthesized by standard reduction with NaBH_4_ in the presence of histidine as capping reagent were used under the same experimental conditions (ESI, Appendix A) [59]. In the presence of TPPS, a further broadening of the plasmonic band was observed over a period of 24 h, together with a drastic decrease of band intensities. This was due to the formation of a dark precipitate, as confirmed by the electronic absorption spectrum recorded the day after, in which an increased baseline was evident (cyano spectrum, Figure 1).

The resonance light scattering (RLS) spectrum (black spectrum, Figure 3) of the Au_10_ clusters in aqueous solution in the absence of porphyrin displayed a Rayleigh scattering profile. After addition of TPPS, a well at 418 nm appeared due to absorption resulting from the presence of porphyrin in its monomeric form (green spectrum, Figure 3).

Upon addition of HCl, the RLS spectrum of the sample showed a peak at 500 nm due to the presence in solution of large ordered J-aggregates, stabilized by a network of electrostatic and solvophobic interactions among porphyrins (blue spectrum, Figure 3). After 24 h, the Rayleigh scattering due to the presence of larger gold structures in solution formed by aggregation phenomena occurring in acidic medium, modulated by the absorbance of the sample, was the only detectable component present (red spectrum, Figure 3).

As the Au_10_ clusters can be stabilized by either l- or d-histidine, in order to confirm the involvement of the chiral capping agent in the aggregated samples, we performed circular dichroism experiments. As previously described, chirality is transferred from the histidine capping agent to the monomeric porphyrin. Upon acidification and aggregation, chirality induced by the two histidine enantiomers was observed, analogously to data already reported for similar systems in the absence of gold nanoparticles [51]. The CD spectra for solutions of both cluster types showed the presence of a weak bisignate Cotton effect, centered in the J-aggregate absorption region (490 nm). d- and l-histidine led to almost mirror image spectra, characterized by a negative and positive Cotton effect (Figure 2). Moreover, a strong light scattering contribution due to the presence of porphyrin aggregates and gold nanostructures in solution broadened the bands.

As already observed for crude AuNCs, after removing the TPPS–AuNP solution from the cuvette, a purple deposit was observed on the quartz surface. The corresponding electronic absorption spectra obtained for both enantiomers showed, in addition to the spectroscopic features of the AuNPs, the typical J-aggregate band at 491 nm (Figure 4).

The RLS signals confirmed the presence of nanostructured material attached to the quartz substrate (Figure 4, inset). The preferential adsorption of the porphyrin J-aggregates on the cuvette surface was further proven by the observation of a residual amount of diacid monomeric TPPS in solution. After transferring the solution into a new cuvette, only the spectrum corresponding to this species could be detected, with the Soret band at 435 nm and with no CD features detectable (ESI, Appendix A). Interestingly, the CD spectra of these films in the J-aggregate spectroscopic region showed an induced bisignate Cotton effect related to the configuration of the amino acid used in the synthesis of the metallic clusters (Figure 5). These dichroic signals cannot be ascribed to linear dichroism, as they do not depend on the cuvette orientation with respect to incident light (ESI, Appendix A). No CD signal was detected for the corresponding AuNP deposit grown from d- and l-histidine (Figure 5, inset).

*Raman and SERS measurements.* The co-deposition of J-aggregated porphyrins and gold nanoparticles prompted us to investigate the possibility of detecting surface-enhanced Raman scattering (SERS) effects. In order to quantify the SERS effect, nitrate was added to the solution and the intensity of the Raman band at 1053 cm^−1^ was taken as an internal standard. No significant spectral changes were introduced due to the presence of the internal standard. Figure 6 shows representative Raman spectra of the J-aggregates obtained with and without Au_10_ clusters for excitation at 514.5 and 785 nm. The spectra of the samples with (Figure 6a,c) and without (Figure 6b,d) Au_10_ clusters were obtained at the same point on the surface of each sample for the two excitation wavelengths. The presence of Au_10_ clusters did not alter the frequency or the relative intensity of the J-aggregate bands, as the spectra were very similar to those previously reported in solution [60]. This result confirms that co-deposition of the J-aggregated porphyrins and the gold nanoparticles generally does not alter their structural integrity, as was previously found for the deposition of the solution phase aggregates onto a gold substrate [46]. However, an intensification of the overall porphyrin spectrum was evident in the samples containing Au compared to those without Au. The effect was considerably more evident in the spectra obtained with 785 nm excitation (Figure 6c,d) than those obtained with 514.5 nm (Figure 6a,b). In fact, in the latter case, the resonance Raman intensification of the porphyrin bands due to the vicinity of the excitation wavelength with the 490 nm aggregate band precluded an accurate evaluation of the SERS effect. In order to quantify the intensification of the porphyrin spectrum, we evaluated the intensity ratio (R) of the band of the internal standard and isolated bands of the aggregate. The spectra were collected on NO_3_^−^ crystals, which displayed both NO_3_^−^ and the J-aggregate Raman signals. Two isolated bands of the porphyrin were used, at 317 and 1232 cm^−1^, corresponding to the out-of-plane vibration involving motion of the pyrrolic hydrogens and to the totally symmetric C_m_-phenyl stretch, respectively [32]. Similar results were also obtained using the 993 cm^−1^ band of SO_4_^2−^, however, in this case, the band of the internal standard overlapped with the band at 986 cm^−1^ of the J-aggregate (data not shown). In the presence of Au_10_ clusters, the J-aggregate spectrum was intensified compared to the 1053 cm^−1^ band of NO_3_^−^.

The SERS intensification is evident from the marked reduction of the R values observed in the presence of Au_10_ (Table 1). The scatter of the R_785_ values for the sample without Au was likely due to variation in crystal size of the internal standard; the bigger/thicker the crystal, the more intense the salt band.

*AFM investigations*. To obtain insight into the structures of the J-aggregates and of the gold entities grown on the glass surfaces, the morphology of the deposited materials was examined by AFM. The samples consisted mainly of small AuNPs, which were aggregates formed by several sub-nanometer AuNCs. Irregular agglomerates, due to clustering of these small objects, can be also detected with average height ca. 60 nm and diameter 130 nm (ESI, Appendix A). These were accompanied by flat and regular objects, probably single AuNCs, of heights in the tens of nm and mean widths of a few hundred nm. In addition, elongated TPPS J-aggregates formed on the glass surface, displaying lengths that ranged between 100 and 1000 nm, 3–4 nm height, and ca. 80 nm width (Figure 7 and ESI, Appendix A). These results are consistent with bundles of collapsed J-aggregates due to the solvent evaporation from the inner compartment of the nanotube [61].

## 4. Conclusions

Nanoassemblies of gold clusters and J-aggregated porphyrins were synthesized and characterized by spectroscopic techniques. The aggregated porphyrin showed a chirality related to the amino acid used in the synthesis of the metal nanostructures. A simple procedure for the deposition of gold nanoparticles on glass substrates was described, which starts with the preparation of easily synthesized and highly reactive Au_10_ clusters. In fact, we found that acidification of aqueous solutions of Au_10_ sub-nanometer clusters led to the formation of plate-like and regular nano-objects on glass surfaces. Our procedure offers the advantage of not requiring the use of expensive and sophisticated equipment and hazardous reagents. The growth of the metallic nanostructures with the co-deposition of TPPS J-aggregates on the substrates was exploited as a test system for the potential use of these nanoparticles in SERS applications. In fact, a significant increase of the Raman signals for the porphyrin J-aggregates deposited on gold plated surfaces was observed compared with J-aggregates deposited on bare glass. Furthermore, these J-aggregates maintained their peculiar CD signals on the glass surface, showing a chirality that is related to the configuration of the amino acid (d- and l-histidine) used in the metal cluster synthesis. This latter observation suggests that the Au_10_ clusters, analogously to other templating systems [62,63], have a role as chiral seeds in the growth of J-aggregates.

## Figures and Tables

**Figure 1 nanomaterials-09-01026-f001:**
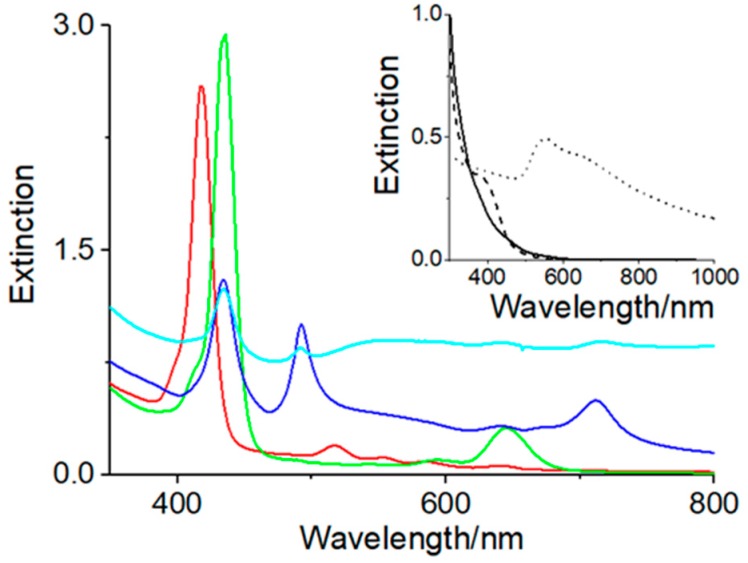
UV–vis spectra of tetrakis(4-sulfonatophenyl)porphyrin (TPPS)–Au_10_ in aqueous solution (red line): instantaneously upon HCl addition (pH 2) (green line), after 1 h (blue line), after 24 h (cyano line). The inset shows the crude Au_10_ NCs for the same experimental conditions in aqueous solution (black full line) upon HCl addition (pH 2) instantaneously (black dashed line) and after 24 h (black dotted line). Experimental conditions: [Au_10_] = 125 μM, [TPPS] = 5 μM.

**Figure 2 nanomaterials-09-01026-f002:**
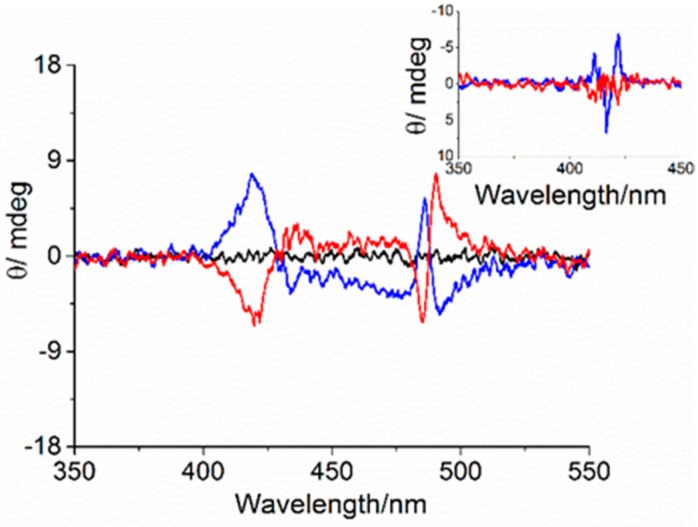
Circular dichroism (CD) spectra of Au_10_ in aqueous solution (black line), Au_10_–TPPS Jagg nanohybrid assemblies for l-(blue line) and d-histidine (red line) before (inset) and 1 h after HCl addition. ([Au_10_] = 125 μM, [TPPS] = 5 μM, pH 2).

**Figure 3 nanomaterials-09-01026-f003:**
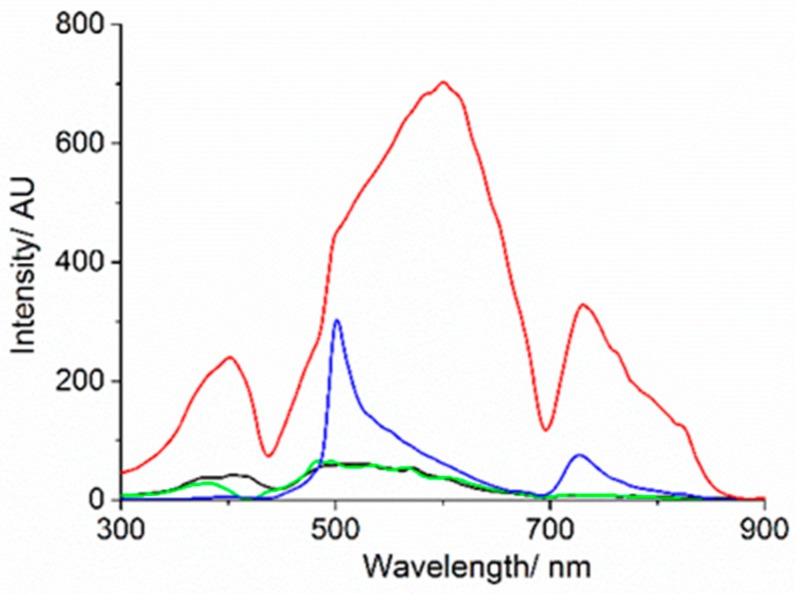
Resonance light scattering (RLS) spectra of Au_10_ in aqueous solution (black line), TPPS@Au_10_ in aqueous solution (green line), upon addition of HCl (pH 2) after 1 h (blue line) and after 24 h (red line). ([Au_10_] = 125 μM, [TPPS] = 5 μM).

**Figure 4 nanomaterials-09-01026-f004:**
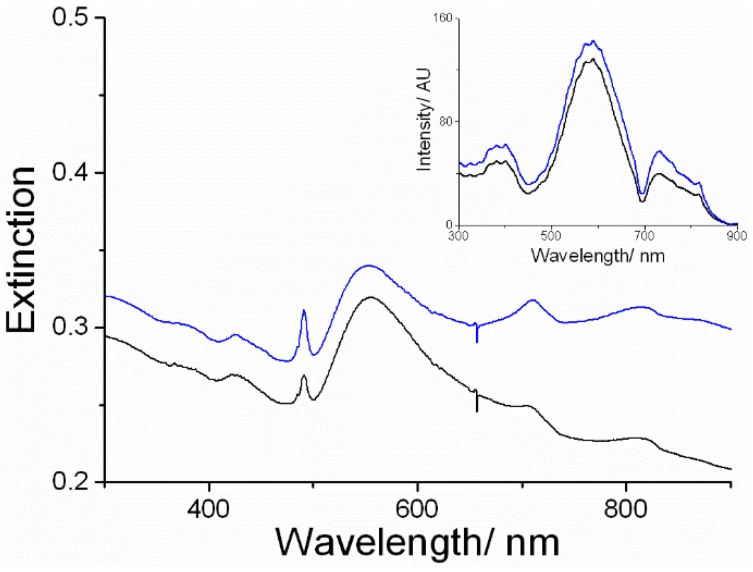
UV–vis spectra of the TPPS@AuNPs deposit left on the cuvette surface, obtained after washing with water, for d- (**black line**) and l- (**blue line**) histidine samples. The inset shows the corresponding RLS spectra.

**Figure 5 nanomaterials-09-01026-f005:**
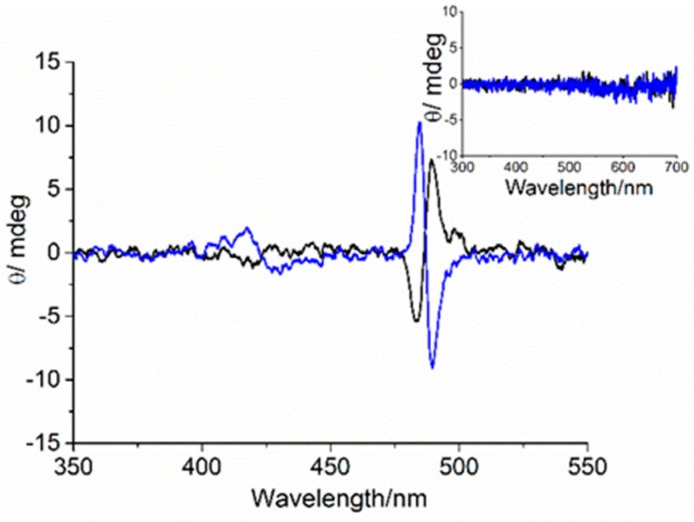
CD spectra of TPPS@AuNP co-deposit for d-(**black line**) and l-(**blue line**) histidine samples. The inset shows the corresponding AuNP deposit grown from d-(**black line**) and l-(b**lue line**) histidine.

**Figure 6 nanomaterials-09-01026-f006:**
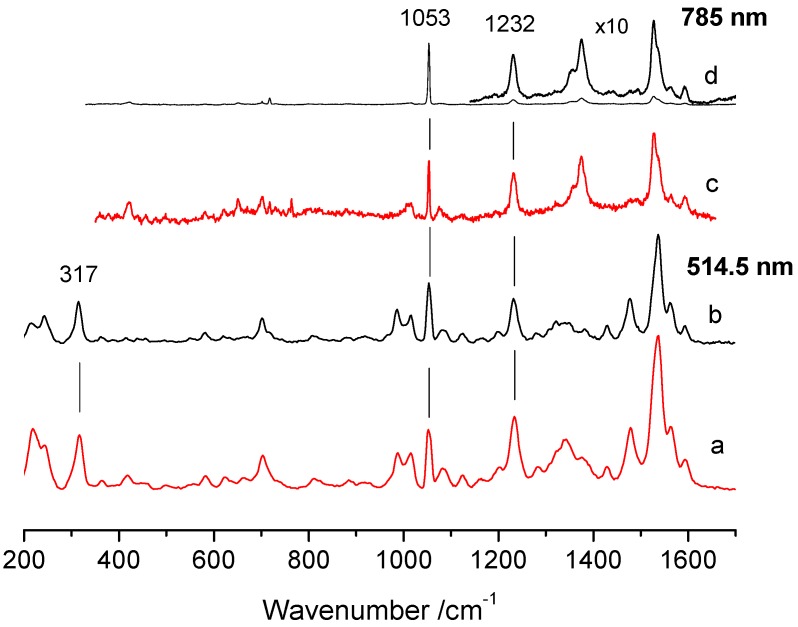
Representative Raman spectra of porphyrin J-aggregates in the presence (**a**,**c**) and absence (**b**,**d**) of Au_10_ clusters with excitation at 514.5 and 785 nm. The spectra of each sample (**a**,**c**) and (**b**,**d**) were obtained at the same point on the sample surface for the two excitation wavelengths. The inset of spectrum (**d**) has been multiplied 10-fold. Experimental conditions: (514.5 nm): (**a**) average of twelve spectra with 10 s integration time; (**b**) average of eleven spectra with 10 s integration time; (785 nm): (**c**) average of seventeen spectra with 30 s integration time; (**d**) average of nine spectra with 30 s integration time. All the spectra have been normalized with respect to the band of nitrate at 1053 cm^−1^.

**Figure 7 nanomaterials-09-01026-f007:**
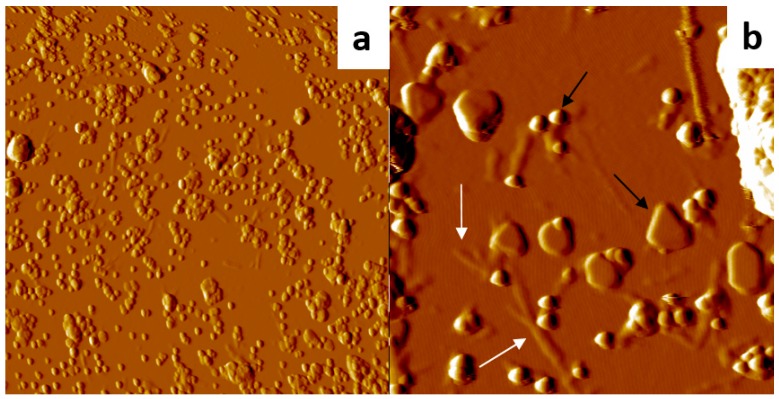
AFM topography images (x gradient) of the deposits formed on a glass substrate immersed in a Au_10_ and TPPS solution at pH 2.0, showing Au nanostructures (dark arrows) with J-aggregates (white arrows). Experimental conditions: (**a**) 9 × 9 μm, Z range = 478 nm; (**b**) 2 × 2 μm, Z range = 145 nm.

**Table 1 nanomaterials-09-01026-t001:** Comparison of the intensity ratio (R) of the 1053 cm^−1^ NO_3_^−^ band and the 317 and 1232 cm^−1^ bands of the J-aggregates obtained with 785 and 514.5 nm excitation.

Point	R_1053/1232_	R_1053/1232_	R_1053/317_
	785 nm	514.5 nm	
Without Au			
9	5.9	0.7	0.9
8	8.9	1.2	1.3
7	34.5	2.4	2.7
6	13.3	1.4	1.4
5	24.3	2.2	2.7
4	16.2	1.4	1.5
With Au			
12	1.0	0.9	1.0
11	1.3	0.9	1.0
10	1.5	0.5	0.7
3	1.3	0.8	1.0
2	2.0	0.5	0.6
1	3.5	2.8	3.1

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
