# Peer review of "Nanohybrid Assemblies of Porphyrin and Au10 Cluster Nanoparticles"

_nanomaterials, 2019, doi:10.3390/nano9071026_

Reviewer 1 Report

The article  “Nanohybrid Assemblies of Porphyrin and Au10  Cluster Nanoparticles “ by Mariachiara Trapani, Maria Angela Castriciano, Andrea Romeo, Giovanna De Luca, Nelson Machado, Barry D. Howes, Giulietta Smulevich, Luigi Monsù Scolarois is devoted to the experimental study of very actual problem of creation of hybrid nanomaterials due to the joint self-organization of inorganic species and  stabilizing organic ligands. The introduction of small metal clusters and nanoparticles in structured organic matrices is of big practical importance due to possible applications of such systems in nanoplasmonics and nanophotonics, surface enhanced Raman spectroscopy and sensing.  The use of chiral stabilizing ligands is of great fundamental interest of the appearance of chirality and optical activity in metal-containing nanosystems.

          The article contains new and interesting experimental results and can be published after minor revisions and clarifying of some points:

 1.     It is not obvious from the article text the exact formation of Au10 clusters, containing just ten gold atoms by borohydride reduction of Au (3+) containing compound in presence of stabilizing agents, Ordinary, the used method led to the formation of the small metal nanoparticles with narrow, but existing size distribution.

2.      In the Figure.7, presenting AFM images of the gold nanoparticles and their aggregates there is no any indication of standard length in horizontal and vertical directions. The sub-nanosized gold clusters couldn’t be detected using such images.

3.     It is not clear the statement on the appearance of the chirality of metallic core of hybrid nanostructure, as also the suggestion that “Au10 clusters have a role as chiral seeds in the growth of porphyrin  J aggregates”.

Author Response

i)      “It is not obvious from the article text the exact formation of Au10 clusters, containing just ten gold atoms by borohydride reduction of Au (3+) containing compound in presence of stabilizing agents, Ordinary, the used method led to the formation of the small metal nanoparticles with narrow, but existing size distribution.”

The synthesis of gold clusters (Au10) has been described in the Materials and Methods section (see p. 3 Gold cluster synthesis) using a method reported in the literature by Yang, X. et al. (ref. 18). This is the system we are investigating in this work. Any reference to gold(III) reduction by NaBH4 in the text was reported to make a comparison with Au10 (see p. 5 ll. 214-217: “It is noteworthy that no metallization of the cuvette wall occurs when Au nanoparticles synthesized by standard reduction with NaBH4 in the presence of histidine, as capping reagent, are used under the same experimental conditions (ESI, Figure S5).[60]”)

ii)    “In the Figure.7, presenting AFM images of the gold nanoparticles and their aggregates there is no any indication of standard length in horizontal and vertical directions. The sub-nanosized gold clusters couldn’t be detected using such images.”

The caption of Figure 7 already describe the actual size of the two images: (a) 9 x 9 mm, Z range = 478 nm; (b) 2 x 2 mm, Z range = 145 nm. As far as the sub-nanosized gold clusters are concerned, we are aware that their sizes is well below the resolution of the AFM image. Actually, this technique has been used to investigate the morphology and sizes of the deposited film upon acidification and we have already pointed out in the main text that the Au10 clusters grow into larger gold nanostructures, together with porphyrin J-aggregates (p. 9, ll. 311-320).

iii)  “It is not clear the statement on the appearance of the chirality of metallic core of hybrid nanostructure, as also the suggestion that “Au10 clusters have a role as chiral seeds in the growth of porphyrin J aggregates”.”

The nature of the optical activity in Au10 clusters is discussed from the viewpoint of the intrinsically chiral core model and the dissymmetric field effect in reference 19. The transfer of chirality from these clusters to the J-aggregates is an experimental observation, strongly related to many examples in the literature on similar phenomena (see e.g. ref. 41, 43, 52). Some new references on the subject have been added (p. 9, l. 340, ref. 64-65)

Reviewer 2 Report

The authors report on the formation of chiral aggregates based on Au10 nanoclusters capped with histidine and TPPS. Overall the work is of interest and can be accepted in Nanomaterials after some revisions.

Several points top to be addressed:

- in Materials and Methods section what means “Au10@Jagg assemblies”?

- the beginning of the Results section is not clear at all! I do not see in Figure 1 any broad band at 320 nm. Please make clearer this discussion, why HCl is added, what is the objective, etc.

- please describe more specifically the interaction between the Au nanoclusters and the TPPS and which kind of chiral aggregates TTPS can form to explain the observed induced CD. Some recent references on the self-assembly and chiroptical properties of chiral porphyrins could be added.

Author Response

i)      “in Materials and Methods section what means “Au10@Jagg assemblies”?”

We used this acronym to indicate the nanoassembly formed by interaction of Au10 and porphyrin J-aggregate. In order to make this point more clear, we added the acronym in the main text on the Introduction section (p. 2, l. 84)

ii)    “the beginning of the Results section is not clear at all! I do not see in Figure 1 any broad band at 320 nm. Please make clearer this discussion, why HCl is added, what is the objective, etc.”

The description of the absorption spectra has been rewritten (p. 3 l.137-139 “The clusters show an absorption edge at around 450 nm that rises very steeply below 320 nm (full line spectrum, inset of Figure 1), together with the typical fluorescence emission centered at 500 nm, as previously reported.[19]”).

As pointed out in the introduction (p. 2 l. 66-68), the formation of TPPS J-aggregates is fostered under acidic conditions. This is also stated in the Results section (p. 5 l 181-183 : “Since histidine Au10 clusters can be successfully prepared over a wide pH range (pH 2–12),[18] we decided to foster porphyrin aggregation by lowering the pH.”).

iii)  “please describe more specifically the interaction between the Au nanoclusters and the TPPS and which kind of chiral aggregates TTPS can form to explain the observed induced CD. Some recent references on the self-assembly and chiroptical properties of chiral porphyrins could be added.”

In the absence of structural data on these nanoaggregates, we consider rather speculative to discuss specifically the interaction between TPPS and Au nanoclusters. Anyway, a discussion on a hypothetical model is already present in the main text (p. 4-5 ll 169-181).

More references on self-assembly and chiroptical properties have been added (p. 9, l. 340, ref. 64-65)